Specificity trumps flexibility—location-based stable associations between Symbiodiniaceae genera and Platygyra verweyi (Scleractinia; Merulinidae)

Keshavmurthy Shashank 1
Tee Hwee Sze 1 2
Kao Kuo-Wei 1 3
Wang Jih-Terng jtwtaiwan@gmail.com jtw@tajen.edu.tw 4
Chen Chaolun Allen cac@gate.sinica.edu.tw 1 3 5
1 Biodiversity Research Center, Academia Sinica , Taipei , Taiwan
2 Taiwan International Graduate Program, International Internship Program, Academia Sinica , Taipei , Taiwan
3 Institute of Oceanography, National Taiwan University , Taipei , Taiwan
4 Department of Oceanography, National Sun Yat-Sen University , Kaoshuing , Taiwan
5 Department of Life Science, Tunghai University , Taichung , Taiwan
Banaszak Anastazia
Electronic publication date: 2020 May 5
Publication date: 2020
Volume: 8
Electronic Location ID: e8791
Received 2019 Sep 25; Accepted 2020 Feb 24
Copyright: ©2020 Keshavmurthy et al.
Copyright year: 2020
Copyright holder: Keshavmurthy et al.
License: This is an open access article distributed under the terms of the Creative Commons Attribution License, which permits unrestricted use, distribution, reproduction and adaptation in any medium and for any purpose provided that it is properly attributed. For attribution, the original author(s), title, publication source (PeerJ) and either DOI or URL of the article must be cited.
License URL: https://creativecommons.org/licenses/by/4.0/

Keywords: Reciprocal transplantation, Thermal histories, Degree heating weeks, Symbiodiniaceae genra shuffling, Nuclear power plant

Funding: Ministry of Science and Technology NSC101-2621-B-001-005-MY3 Academia Sinica Thematic Grants AS-104-SS-A03 The study was supported by grants from the Ministry of Science and Technology (NSC101-2621-B-001-005-MY3) and Academia Sinica Thematic Grants (AS-104-SS-A03). The funders had no role in study design, data collection and analysis, decision to publish, or preparation of the manuscript.

==============================
This study monitored symbiont communities bi-monthly in native coral cores used in a reciprocal transplantation of the coral Platygyra verweyi over two years (2014–2016) and samples of mother colonies from three locations with variable thermal regimes; our results show that associating with multiple Symbiodiniaceae genera (Cladocopium spp. and Durusdinium spp.) is not a prerequisite for symbiont shuffling. Platygyra verweyi associates with certain Symbiodiniaceae genera based on location. Results of quantitative real-time PCR indicated small-scale temporal changes in Symbiodiniaceae genera compositions from 2014 to 2016; however, these changes were not enough to invoke shuffling or switching, despite degree heating weeks exceeding 6 °C-weeks in 2014 and 4 °C-weeks in 2015, which usually resulted in substantial coral bleaching. Microsatellite analysis of the P. verweyi host showed no genetic differences among the study locations. Our results suggest that P. verweyi undergoes long-term acclimatization and/or adaptation based on microgeographic and local environmental conditionsby altering its combinations of associated Symbiodiniaceae. Results also suggest that shuffling might not be as common a phenomenon as it has been given credit for; corals thrive through specific associations, and many corals could still be vulnerable to climate change-induced stress, despite being promiscuous or able to associate with rare and background Symbiodiniaceae genera.

Background

Corals in reefs around the world have been facing rapid declines in health over the past several decades due to increased and prolonged occurrences of climate change-induced seawater temperature anomalies, which often pass their threshold limits (IPCC, 2018; Hughes et al., 2017; Hughes et al., 2018). A common manifestation of this stress is coral bleaching due to the breakdown of coral-Symbiodiniaceae associations (Brown, 1997). Moreover, corals have the potential to acclimate to climate change-induced stressors—over a short period of time (single generation)—through phenotypic plasticity or associating with specific combinations of stress resistant Symbiodiniaceae genera through natural selection; this may be the overriding determinant of their survival (Marshall & Baird, 2000). However, a beneficial association between a coral host and Symbiodiniaceae is a complex and holistic process that depends on whether the relationship that the coral host has with the Symbiodiniaceae genera is specific or flexible  (Baker, 2003; Lajeunesse et al., 2010; Silverstein, Correa & Baker, 2012). Corals are known to associate with a wide range of Symbiodiniaceae genera. There are nine genera of Symbiodiniaceae, and each has its own characteristic traits that help its coral host survive in a wide range of environmental niches (Lajeunesse et al., 2018). Studies have shown that symbiosis between coral hosts and different Symbiodiniaceae genera contributes to the divergence in coral thermal tolerance under different environmental conditions (Lajeunesse et al., 2010; Weber & Medina, 2012). For instance, species of Durusdinium are considered to be heat tolerant (Baker, 2003; Jones et al., 2008; Sampayo et al., 2008; Ulstrup & Oppen, 2003); most species Cladocopium are stress sensitive but several are relatively stress tolerant (e.g., in-hospite Cladocopium C15, Cladocopium thermophilum) (Fisher, Malme & Dove, 2012; Hume et al., 2015). Durusdinium-associated corals are also known to inhabit reef environments that experience large fluctuations in surface seawater temperature (Lajeunesse et al., 2010; Lien et al., 2007; Ghavam Mostafavi et al., 2007) and be more resilient to heat-treatment experiments (Oliver & Palumbi, 2011).

One widely-known mechanism that helps some corals acclimate to stressful environments is shuffling and/or switching their associated Symbiodiniaceae genera (Baker et al., 2004; Berkelmans & Van Oppen, 2006; Sampayo et al., 2008; Jones & Berkelmans, 2008; Silverstein, Cunning & Baker, 2015; Cunning, Silverstein & Baker, 2015; Boulotte et al., 2016). It has been proposed that coral hosts adjust to increasing seawater temperatures using switching—which involves existing Symbiodinium being expelled and replaced by novel Symbiodinium from the environment—and shuffling between stress-sensitive (generally Cladocopium sp.) and stress-resistant types (generally Durusdinium sp.) in the existing symbiont communities  (Baker et al., 2004; Berkelmans & Van Oppen, 2006; Buddemeier, Fautin & Ware, 1997; Baker, 2001). Shuffling between Symbiodiniaceae genera has been found to benefit some coral species (Jones et al., 2008; Sampayo et al., 2008) because increasing the abundance of stress-tolerant Symbiodiniaceae genera in a multi-symbiont association helps corals withstand above-threshold seawater temperature anomalies (Lien et al., 2007; Ghavam Mostafavi et al., 2007; Oliver & Palumbi, 2011). A transition from thermally sensitive to tolerant dominant symbionts can increase the likelihood that corals survive thermally-induced bleaching (Bay et al., 2016). However, a later study (Goulet, 2006) argued that not all corals can change their symbionts, because the mechanism of shuffling requires that a coral species hosts multiple Symbiodiniaceae genera (at least one stress tolerant and one stress resistant). Some coral species have the ability to fluctuate between Symbiodiniaceae genera on a temporal scale (e.g., Hsu et al., 2012). The occurrence of multiple Symbiodiniaceae genera at low densities might lead to either shuffling or switching to beneficial Symbiodiniaceae genera over time (Hsu et al., 2012). In some cases, the coral host may revert back to its original composition of either a single dominant Symbiodiniaceae species or multiple species/genera (see Thornhill et al., 2006). However, there are also many cases in which the host maintains stable symbiosis with a particular Symbiodiniaceae genus, irrespective of environmental perturbations (see Thornhill et al., 2006; Thornhill, Fitt & Schmidt, 2006). This acclimatization mechanism, although limited, may help corals survive the effects of ocean warming in the near future (Berkelmans & Van Oppen, 2006; Palumbi et al., 2014).

In order to assess whether associating with multiple Symbiodiniaceae is in itself enough for shuffling to happen, we analysed coral samples collected over time from locations with different thermal regimes to determine how location influences corals’ abilities to shuffle and subsequently survive. Nanwan, Kenting National Park, Taiwan is a reef site located southwest of the third Nuclear Power Plant Outlet (NPP-OL) that has been affected by the continuous discharge of thermal effluent flowing directly into the existing coral community as a result of the near-shore current (Chiou, Cheng & Ou, 1993) since the power plant opened in 1984. NPP-OL has seawater temperatures similar to those predicted for oceans around the world by 2050 (IPCC, 2018), making it an ideal location to conduct studies related to the effects of climate change. NPP-OL has significantly different community compositions and settlement patterns compared to other sites such as the Nuclear Power Plant Inlet (NPP-IL)  (Chou et al., 2004), suggesting that the thermal effluent has had a great impact on its benthic invertebrate and fish communities. Corals present at NPP-OL have also experienced several bleaching events over time (see Fan, 1991; Hung & Huang, 1998). The coral communities in the shallow water (3 m) are dominated by thermally tolerant symbiont types (Hsu et al., 2012; Keshavmurthy et al., 2012; Keshavmurthy et al., 2014). The increasing prevalence of stress-tolerant Durusdinium sp. at reef sites closer to NPP-OL reflects the consequences of long-term thermal effects; this prevalence also makes NPP-OL an ideal site to study holobiont dynamics under thermal stress, although other physical differences (temperature fluctuations, upwelling, and internal waves; see Keshavmurthy et al., 2019) between NPP-OL and other sites could also be involved (Hsu et al., 2012; Keshavmurthy et al., 2012; Keshavmurthy et al., 2014; Lee et al., 1997; Lee, Chao & Fan, 1999).

In this study, we collected data from in situ reciprocal transplant experiments (RTE) on nubbins of P. verweyi collected from NPP-OL, NPP-IL, and Wanlitung (WLT) in Kenting National Park (KNP) between 2014 and 2016 (see Kao et al., 2018). Using the samples from each location of origin (tagged mother colonies and native cores used in the transplant experiment), we tested whether corals in their native environments will undergo shuffling over time and associate with favourable Symbiodiniaceae genera.

Methods

Samples used in the experiment were collected with permission from the Kenting National Park headquarters (permit numbers 1040008112 and 1040002080).

Study area and coral species

All experiments and sampling were carried out at three locations: Nuclear Power Plant Outlet (NPP-OL), Nuclear Power Plant Inlet (NPP-IL), and Wanlitung (WLT) in Nanwan, southern Taiwan (Fig. 1). NPP-OL (21°55′54.4″N, 120°44′42.7″E) and NPP-IL (21°57′20.3″N, 120°45′14.2″E) are located within Nanwan in the Kenting National Park (KNP), Taiwan, and WLT (21°59′41.0″N 120°42′19.6″E) is located on the west coast of KNP, approximately 12 km from the nuclear power plant area. Due to the tidally-induced upwelling in Nanwan (Lee et al., 1997), the maximum daily seawater temperature fluctuation at NPP-OL and NPP-IL can exceed 8 °C in the summer season. In this study, NPP-IL and WLT were both included to assess the potential role of thermal variability on P. verweyi facing prolonged thermal stress. The massive coral species P. verweyi generally occurs in shallow water (2–4 m) and was found to associate with the Symbiodiniaceae genera Cladocopium sp. and/or Durusdinium sp. in KNP (Keshavmurthy et al., 2012).

Figure 1 Map showing the reciprocal transplant experiment locations in the Kenting National Park, Taiwan.

NPP-OL, Nuclear Power Plant Outlet; NPP-IL, Nuclear Power Plant Inlet and WLT-Wanlitung. Source credit: Keshavmurthy et al. (2012).

Coral samples

Two types of samples were used in the experiment: the “mother colonies” were the original colonies (from the 2014–2016 transplantation experiment), and the “native cores” were samples cut from a mother colony and kept at the same location.

Tagged mother colonies at each location (five from NPP-IL in 2014, eight each from NPP-OL and WLT from 2014 to 2016) were sampled approximately every two months throughout the experimental period. During each sampling, a piece (2 cm diameter core) of coral was cut and fixed in 95% Ethanol for DNA extraction and qPCR analysis.

Samples were taken from the coral cores in the racks used in the transplantation experiment (see Kao et al., 2018 for detailed explanation of the transplantation experiment). All samples were taken from the set of control racks installed at each location from the reciprocal transplant experiment, and hence referred to as “native cores”. Five native cores (one from each colony) from each rack in 2014 (at all three sites) and one piece (2 cm in diameter) from each of the 30 cores in 2015 (from the racks at NPP-OL and WLT) were sampled at each location. Sampling for the 2014 set was carried out in April and September of 2014 and January, March, May, July, September, and November of 2015. Sampling for the 2015 set was carried out in April, September, and November of 2015 and January and April of 2016. All samples were fixed in 95% Ethanol for DNA extraction and qPCR analysis.

Seawater temperature

The seawater temperature was recorded in situ at 30-minute intervals using data loggers (HOBO; Pendant™, USA) deployed underwater near the transplant racks (1–2 m) at each study site. The raw temperature data were transformed into Degree Heating Weeks (DHW) (Wellington et al., 2006; Liu, Strong & Skirving, 2003) to assess both the intensity and duration of the thermal stress for each experiment group. Although this indicator is typically used to monitor large-scale bleaching, it was also used in another study to assess the cumulative thermal stress on heat-treated corals, but daily (Schoepf et al., 2015). DHW was calculated as follows. First, the weekly mean temperature for each study site was calculated from raw temperature data. Second, the maximum of the monthly mean temperatures (MMM; NPP-OL = 29.63 °C; WLT = 29.28 °C; NPP-IL = 28.37 °C) was obtained from data loggers (HOBO; Pendant™, USA) deployed at each study location. Finally, the weekly mean temperature was subtracted from MMM to determine temperature anomalies; only temperatures at least 1.0 °C above the MMM within the previous 12 weeks were considered anomalies and summed to obtain the DHW. The conceptual calculation equation is listed below

DHWTWO= ∑TNPP-OL−MMMWLT≥1°C

where TNPP-OL is the weekly mean temperature at NPP-OL, MMMWLT is the MMM of WLT, etc.

The projections of DHW in Nanwan were obtained from Palumbi et al. (2014) using a 1° × 1° resolution grid of reef cells located in southern Taiwan.

DNA extraction

DNA extraction was carried out using a salting-out method modified from  Ferrara et al. (2006). Coral tissue was lysed overnight in a 2-mL Eppendorf tube with 200 µL of lysis buffer [0.25 M Tris, 0.05 M EDTA at pH 8.0, 2% sodium dodecylsulfate (SDS) and 0.1 M NaCl] and 10 µL of 10 mg/mL proteinase E at 55 °C in a water bath. NaCl (210 µL at 7 M) was added to the lysed tissue in the tube, and the sample was mixed by carefully inverting the tube. The solution was then transferred to a 2-mL collection tube containing a DNA spin column (Viogene, USA) and centrifuged at 8000 rpm for 1 min. The lysate was washed twice with 500 µL of ethanol (70%) by centrifuging at 8,000 rpm for 1 min at each step, with an additional centrifugation step at 8000 rpm for 3 min to dry the spin column. The column was dried further at 37 °C for 15 min, then the DNA was eluted with 50 µL of preheated (65 °C) 1X TE buffer, with a final centrifugation at 15000 g for 3 min. The quality of genomic DNA was checked using a 1% agarose gel. The concentrations of genomic DNA were determined using NanoDrop 2000 (Thermal Scientific, USA).

Real-time quantitative PCR

The copy numbers of Cladocopium sp. and Durusdinium sp. in P. verweyi samples were determined under a LightCycler® 480 Instrument II (Roche, Switzerland) using a protocol modified from Mieog et al. (2007). Each 10 µL qPCR reaction consisted of 5 µL 1x SYBR Fast Master Mix, 0.5 µL UF primer (2 nM/µL), 0.5 µL CR or DR primer (2 nM/µL), 7.5 µL ddH2O, and 2.5 µL DNA templates (equal to 1 ng of genomic DNA). The following primer sets were used: ITS1 clade C-specific reverse primer (CR) 5-AAGCATCCCTCACAGCCAAA-3, clade D-specific reverse primer (DR) 5-CACCGTAGTGGTTCACGTGTAATAG-3, and universal forward primer (UF) 5-AAGGAGAAGTCGTAACAAGGTTTCC-3 (Sampayo et al., 2008). Each sample was run in triplicate (technical replicates), as was a no-template control (NTC) with ddH2O. Plasmid standard curves were run in duplicate with P. verweyi samples to quantify the copy numbers of each symbiont. Plasmid standard curves were generated using PCR products from Cladocopium sp. and Durusdinium sp., which were ligated into pGem®-T Easy vectors (Promega, USA), transformed, and amplified using E. coli. Copy numbers of final products were calculated by first quantifying the concentration of plasmid DNA through NanoDrop 2000 (Thermal Scientific, USA), then dividing by the mass of the plasmid [mass of each plasmid copy = 3015 bp (vector) + 100 bp (inserted PCR product) × 1.096e−21g/bp = 3.4 × 10−18 g], and finally multiplying by the plasmid DNA template volume (2.5 µL) in each reaction. Serial dilutions of 1:10 from 3 × 106 and 30 copies of the plasmid standard containing Cladocopium sp. and Durusdinium sp. sequences, respectively, were generated in the end. The qPCR cycling settings were: 40 two-step cycles of 15 s at 95 °C and 1 min at 60 °C. Melting curves were generated by starting at 60 °C and increasing the temperature with a ramp speed of 0.11 °C/s until it reached 95 °C. Fluorescence data were collected after each annealing step, and five readings were collected every second during the melting curve analysis. Crossing points (Cp) were determined by Light Cycler 480 software version 1.5 (Roche, Switzerland) using the second derivative method, which represents the cycles with the maximum number of fluorescence signals in each sample (Rasmussen, 2001). Samples with Cp values that varied from the other two technical replicates by 1 were excluded from analysis. Samples were re-run if all Cp values of technical replicates varied from one another by 1. Since high variation occurred in Cp values (varied more than 1) within technical replicates of each sample when Cp > 34, the cut-off cycle was set to 34 to avoid false positives caused by the formation of non-specific fluorescence. Average copy numbers of symbiont clades C and D were obtained individually, and the formula for determining relative symbiont abundance in this study is listed below, following a correction suggested by Mieog et al. (2007): (Clade D copy numbers/3)/[(Clade D copy numbers/3) + Clade C copy numbers].

Population genetics analysis of P. verweyi

To demonstrate the presence of a genetic structure in P. verweyi at NPP-OL and WLT, eight polymorphic microsatellite loci were used to examine 30 P. verweyi colonies (including transplanted colonies) found within two transplanted sites. Three published microsatellite tetramer markers developed from P. sinensis  (Tay et al., 2014) and P. daedalea (Miller & Howard, 2004) were used. Another five microsatellite dimer markers specific to P. verweyi were developed by Next Generation Sequencing approaches (Yang et al., 2018). Eight microsatellite markers were amplified following the effective universal fluorescent labeling method (Schuelke, 2000). Amplifications performed using 25 µL reactions contained 10 ng of DNA template, 1X of VeraSeq Buffer II (Qiagen Beverly, USA), 0.5U of VeraSeq 2.0 high-fidelity DNA polymerase (Qiagen Beverly, USA), 0.2 mM of dNTP mix, 0.08 µM of specific forward primer-attached M13 (-21) tail (5′-TGT AAA ACG ACG GCC AGT- 3′) (18 bp), 0.2 µM of specific reverse primer, and 0.2 µM TAMRA-labelled universal M13 (-21) primer (5′-TGT AAA ACG ACG GCC AGT- 3′) (Schuelke, 2000). The PCR conditions were: 1 cycle at 98 °C for 30 s; 25 cycles of 98 °C for 10 s, specific primer annealing temperature (Table 1) for 30 s, and 72 °C for 30 s; followed by 10 cycles of fluorescent-labelled M13 amplification: 98 °C for 10 s, 53 °C for 30 s, 72 °C for 30 s; and a final elongation of 10 min at 72 °C. For microsatellite genotyping, samples were electrophoresed on 5% urea denaturing polyacrylamide gels using the Gel-Scan 3000™ real-time DNA fragment analysis gel Electrophoresis System (Corbett Robotics, Australia). Allele size was detected by the software Gene Profiler 4.05 (Scanalytics) with the internal lane size standard (GeneScan™-350 TAMRA™, Applied Biosystems). Characteristics of microsatellite loci—such as number of alleles and mean observed and expected heterozygosities—were calculated using GenAlEx v.6.502 (Peakall & Smouse, 2012). Genepop was used on the web to test for linkage disequilibrium and significant departure from Hardy-Weinberg equilibrium (HWE). None of the loci showed HWE deviation or linkage disequilibrium after Bonferroni correction (Rice, 1989). Population differentiation was inferred using ARLEQUIN v3.5 (Excoffier & Resources, 2010). Inference population genetic structure was estimated using a Bayesian clustering approach implemented in STRUCTURE v.2.3.4 (Pritchard, Stephens & Donnelly, 2000). The admixture model and allele frequency correlation were used. Values of number of genetic clusters (K) from 1 to 2 were tested by running three replicate simulations per K with 1,000,000 Markov chain Monte Carlo repetitions and 100,000 burn-in iterations.

Table 1 Characteristics of eight microsatellite loci for 60 colonies of P. verweyi collected at NPP-OL and Wanlitung.

Locus	Primers sequences	Repeats	Size of alleles	Tm (°C)	No. alleles	HE	
PV9	F:aCAACTTAAATGGTATCATCGTG	(AG)26	140–168	50	9	0.85	
R: GTGCCCTATTTTATGTGACAA	
PV19	F:aTAGTCAGTGGCATCTGAGAGT	(TG)30	161–197	53	16	0.83	
R: CTCATTTCCTCCTAAGCTTTC	
PV22	F:aTCACTTGCTATAACCTTCTCCT	(TG)12	140–164	50	10	0.86	
R:TCCACCTCTCCAACTAGTTATC	
PV56	F:aTTGACTCGTCAATCACCTATC	(TC)14	144–166	50	6	0.75	
R: GCTAGCACTGATCAAACGAT	
PV57	F:aACAGACAGAGACAGACAGAACA	(TC)14	103–119	50	6	0.60	
R: CAGTTCACCTGTCCATTTG	
Plsi4.02	F:aACAATTCGGATATGTAGC	(AAAC)11	136–170	50	14	0.85	
R:GTTTCTTTGGTTTGGTTTGTTCTC	
Plsi4.24	F:aTTATCTTGGTTCAGACAGACAG	(ACAG)10	126–158	59	9	0.77	
R:GTTTGACAACTCTAATGAAGGTCAG	
PD31	F:aGACAAGTAATGTGTAAATCGTTGTCC	(CCAT)7	156	56	7	0.54	
R:aCTGTTAGAGTATCATGTCCTGAAGC	
Notes.

a primer attached with M13 (-21) tail (5′-TGT AAA ACG ACG GCC AGT-3′) (18 bp).

Statistical analysis

All statistical analyses in this study were performed in R version 3.1.1. (R Core Team, 2014). Differences in daily mean seawater temperatures and daily seawater temperature fluctuations between sites were tested using Kruskal-Wallis test followed by Dunn’s post hoc test with Bonferroni adjusted p-values.

Results

Seawater temperature

Weekly average seawater temperatures were plotted from the data collected by data loggers from 2014 and 2106. Both monthly and daily average seawater temperatures at NPP-OL were significantly higher (2.0–3.0 °C) than at adjacent locations (Fig. 2A). In 2014, the average summer (June to August) daily seawater temperature at NPP-OL (30.11 ± 1.07 SD °C) was different from those at WLT (29.59 ± 0.67 SD °C; Dunn’s post hoc test, p < 0.001) and NPP-IL (28.61 ± 0.96 SD °C; p < 0.001). In 2015, the average summer daily seawater temperature at NPP-OL (29.77 ± 1.12 SD °C) was different from that at WLT (29.52 ± 0.52 SD °C; Wilcoxon rank sum test, W = 5097, p < 0.05). The daily seawater temperature fluctuation at NPP-OL (2.38 ± 1.04 SD °C) was different from that at WLT (1.57 ± 0.70 SD °C) (Wilcoxon rank sum test, W = 655557, p < 0.001). The heating event (≥ 30 °C) at NPP-OL occurred for a longer time each day than at WLT.

Figure 2 Seawater temperature (weekly average) trend (A) and DHW (B) at three locations: Red line—NPP-OL, Black line—WLT and Blue line—NPP-IL, in 2014 and 2015.

Typhoons in 2015 are represented as arrows.

The daily seawater temperature fluctuation at NPP-OL (2.23 ± 1.00 SD °C) was different from those at WLT (1.53 ± 0.58 SD °C; p < 0.001) and NPP-IL (1.80 ± 1.30 SD °C; p < 0.001), while the fluctuations did not vary between WLT and NPP-IL (p = 1.000). During the summer, however, the daily seawater temperature fluctuation at both NPP-OL and NPP-IL was more than 7 °C (maximum 9.12 °C at NPP-OL and 7.19 °C at NPP-IL). The daily heating event (≥30 °C) occurred for the longest time at NPP-OL.

In 2014, repeated seawater temperature anomalies (the weekly mean seawater temperature exceeding the bleaching threshold) occurring at NPP-OL during the summer, resulting in a DHW of 6.4 °C-weeks, while those at NPP-IL and WLT were 2.4 and 1.0 °C-weeks respectively (Fig. 2B). DHW greater than 4.0 °C-weeks results in a NOAA Alert Level 1, meaning that bleaching is likely. The DHWs started to decrease gradually in the fall. In 2015, the DHWs for both NPP-OL and WLT were below the threshold limit of 4.0 °C-weeks—3.8 and 1.0 for NPP-OL and WLT, respectively.

Temporal variation in Symbiodiniaceae genera associated with P. verweyi at NPP-OL

The results of the real-time qPCR analysis of the samples from the 2014 experiment (Fig. 3A) indicated that, until January 2015, the symbiont communities in the native cores of NPP-OL were dominated by Durusdinium spp. In March 2015, the relative proportions of symbionts changed: Cladocopium spp. were present at various percentages (range: 1–21%; mean: 6%; Fig. 3A), and Durusdinium spp. accounted for the rest. Symbiont dynamics in the core with 21% Cladocopium spp. did continue to fluctuate (13% in May; 18% in July and September; Fig. 4). However, this core was dead by November 2015. All remaining cores survived, fluctuating between 1–5% (mean: 2%) Cladocopium spp. for the entire experiment.

Figure 3 Symbiodiniaceae genera trends (A) in five native cores of P. verweyi during the 2014 transplantation experiment and (B) in 30 native cores of P. verweyi during the 2015 transplantation experiment.

Each colour block represents one core sampled from one individual colony. In the case of NPP-IL, no samples were collected after July 2015. X = the cores were dead.

Figure 4 Symbiodiniaceae genera trends in bi-monthly sampling of tagged mother colonies at each location between 2014 and 2016.

Each colour block represents one core sampled from one individual colony. In the case of NPP-IL, there were no samples after November 2015.

In the 2015 experiment (Fig. 3B), the symbiont community in the native core samples from NPP-OL were dominated by Durusdinium spp. However, qPCR suggested that the cores in the samples from April 2016 were associated with Cladocopium spp. in addition to the already present Durusdinium spp. Previously undetected levels of Cladocopium spp. increased enough to be detected by qPCR. Cores were found to contain 3–61% (mean: 22.6%) Cladocopium spp. in April 2016.

Bimonthly sampling of the mother colonies revealed that P. verweyi was specific with respect to which Symbiodiniaceae genera it associated with (Fig. 4). Colonies sampled from NPP-OL showed changes in associated Symbiodiniaceae, meaning that none of the 30 colonies analysed were associated with 100% Durusdinium spp. throughout the entire study period. However, the fluctuation between Durusdinium spp. and Cladocopium spp. was not large. Colonies were found to host 91–100% Durusdinium spp. (seven of the nine colonies)—one colony hosted 50 and 80% at two sampling times—with the exception of one colony, which associated with 72 and 99% Durusdinium spp. in November 2014 and January 2015, respectively. However, this colony changed its relative proportion of symbionts to 100% Cladocopium spp. in March 2015, and then reverted to 97% Durusdinium spp. in May 2015. This was the only colony that had a large fluctuation in associated Symbiodiniaceae between the two Symbiodiniaceae genera.

Temporal variation in Symbiodiniaceae genera associated with P. verweyi at WLT

In the 2014 experiment samples from WLT, a predominance of Cladocopium spp. was observed in the majority of the cores (Fig. 3A), with the exceptions of two cores found dead from March 2015 onwards and one colony that reverted to Durusdinium spp. (83%).

In 2015, cores from WLT were mainly associated with Cladocopium spp., ranging from 5–99% (mean: 31%) (Fig. 3B). Five of the 30 cores were associated with Durusdinium spp. throughout the experimental period. However, in November 2015, 12 of the 30 cores were associated with Durusdinium spp. (range: 1–87%; mean: 32%). Three cores at WLT that were mainly associated with Cladocopium spp. were dead by the end of the sampling period in April 2016 (Fig. 3B).

In the case of the WLT bimonthly analysis of the samples from the mother colonies, seven of the eight colonies analysed were associated only with Cladocopium spp., except for one colony, which also associated with Durusdinium spp. (2% in November, 2014; 1% in July, 2015; and 2% in April, 2016) in addition to Cladocopium spp. (Fig. 4).

Temporal variation in Symbiodiniaceae genera associated with P. verweyi at NPP-IL

In 2014, all the cores at NPP-IL were found to be predominantly associated with Cladocopium spp. throughout the experimental period. None of the cores were associated with Durusdinium spp. (Fig. 3A).

With respect to the bimonthly analysis of the samples collected from the mother colonies, again all the samples were predominantly associated with Cladocopium spp. throughout the sampling period. However, some colonies did show the presence of Durusdinium spp. (1% in March, May, and November; 29% in October, 2015; Fig. 4).

Microsatellite analysis of host samples from NPP-OL and WLT

Microsatellite analysis revealed that a total 60 P. verweyi colonies (30 each from NPP-OL and WLT) exhibited six to 16 alleles per locus for all eight microsatellite loci, with a mean expected heterozygosity of 0.756 ± 0.122 (Table 1). Average gene diversity of P. verweyi at NPP-OL across eight loci was 0.608 ± 0.393; the average for populations at WLT was 0.657 ± 0.460. The pairwise genetic differentiation Fst value between the two sites was -0.00814, and the p-value showed no significant difference. The genetic structure analysis by Bayesian clustering between NPP-OL and WLT also showed no significant differences and no genetic isolation between the two locations (Fig. 5).

Figure 5 Bar plot of STRUCTURE.

Bayesian clustering analysis for eight loci genotypes among the NPP-OL and WLT P. verweyi populations. This bar plot assumed the number of population K = 2. 1,000,000 times MCMC runs.

Discussion

The present study shows spatial variation but specificity in the dominant Symbiodiniaceae genera in the coral P. verweyi, and this may be related to local thermal histories. What was seen in P. verweyi is an almost stable association with a dominant Symbiodiniaceae genus at each sampling time from 2014 to 2016. A fluctuation was observed between Symbiodiniaceae in native cores and/or mother colonies, but this was not a general phenomenon. There was a certain level of temporal and spatial fluctuation in the native coral cores and tagged mother colonies. For example, in the 2014 experiment, some cores at NPP-OL (nubbins here are associated with Durusdinium spp.) did acquire low percentages of Cladocopium spp. (in March 2015) relative to the already present Durusdinium spp. (Fig. 3A). A similar fluctuation was seen in one core from WLT, which was found to have acquired up to 83% Durusdinium spp. However, none of the cores at NPP-IL showed any fluctuation (Fig. 3A). A similar pattern was observed in the tagged mother colonies, with low levels of fluctuation and Symbiodiniaceae genera acquisition (Fig. 4).

Considering the differences in seawater temperature regimes among the three locations (Fig. 2A), this study assumed that native cores and mother colonies would show different levels of Symbiodiniaceae genera shuffling through time. However, the results showed that one Symbiodiniaceae genus was always dominant in each sample (Figs. 3 and 4). For example, samples at NPP-OL were always dominated by Durusdinium spp. Such preference could be because corals at NPP-OL are exposed to long-term seawater temperature stress at shallow depths (1–5 m) and hence are naturally acclimatized to an association with Durusdinium spp.

Association with a particular dominant symbiont could help P. verweyi dominate shallows at various locations in KNP. During the experiment and sampling periods (2014–2016), there was one major bleaching event (2014) and several typhoons (2015). Irrespective of the type of symbiont P. verweyi associated with, none of the corals experienced any clear bleaching—although some degree of paling of the tissue was observed. Increased seawater temperatures in 2014 resulted in DHWs of 6, 2.5, and 1 °C-weeks at NPP-OL, WLT, and NPP-IL, respectively (Kao et al., 2018). DHW in 2015, however, was below 4 °C-weeks in both NPP-OL and WLT (Fig. 2B). The lower DHW values in 2015 could be attributed to the typhoons that occurred that year: the southern coast of Taiwan was hit by three typhoons in the summer, July–September 2015, resulting in the seawater temperature cooling. Such intense changes in seawater temperature conditions might have resulted in corresponding fluctuations in associated Symbiodiniaceae genera in some cores or tagged mother colonies (Figs. 3 and 4). Temperature anomalies found in 2014, with high DHWs, were not enough to yield any pronounced shuffling or switching in this coral.

Our observations beg the following questions: how common is shuffling in corals that can associate with two different symbionts, and does flexibility in symbiont associations via shuffling aid corals under stress? Studies have pointed out that corals often associate with two or more symbionts, with one being dominant and others present at low proportions (<5%, background symbiont) (Silverstein, Correa & Baker, 2012). Therefore, corals have potential to shuffle by regulating their proportions of background symbionts when faced with unfavourable conditions. In this study, we detected symbionts at abundances as low <1%; however, such low concentrations might not be a prerequisite for shuffling or even switching. Also, the recent use of NGS amplicon sequencing has uncovered a rare biosphere with the potential to shuffle and/or switch between different Symbiodiniaceae genera. For example,  Thornhill et al. (2006) investigated the Symbiodiniaceae rare biosphere in two Pocilloporid species from Lord Howe Island in the Great Barrier Reef over two years. Their results showed that, following two consecutive bleaching events, the species shuffled and became associated with new Symbiodiniaceae genera (most <1% of the relative abundance, with one resistant type reaching 33% of the relative abundance).

On the other hand, Bay et al. (2016) suggested that a pre-stress Symbiodiniaceae (D:C) ratio of <0.003 limits the ability of corals to survive bleaching after shuffling. And another study (Lee et al., 2016) showed that variation in the presence and abundances of background or low percentages of symbionts in corals is not necessarily related to shuffling, and may have little or no importance in coral physiology. Future studies need to examine the physiological role of those rare biospheres in terms of supporting corals’ responses to stress.

It may be argued that specificity to a particular Symbiodiniaceae genus depends on the host. We performed genetic analyses on the host using mitochondrial and nuclear markers in a previous study (Keshavmurthy et al., 2012) and microsatellite markers in this study. The results showed no genetic difference in the host between locations. This suggests that other factors influence the type of association we see in this coral, including possible local and microgeographic adaptations to seawater temperature. It may be that using more advanced techniques would help uncover prevalent genetic differences between hosts from two locations in KNP, as was shown in a recent study— (Howells et al., 2016), which showed a clear genetic difference in P. daedalea- associated Symbiodiniaceae between Oman and Abu Dhabi in the Persian Gulf, and hence demonstrated a difference in their eco-physiological behaviour.

Both macro- and micro-environmental differences between locations could dictate Symbiodiniaceae genera associations in P. verweyi. For example, typhoons and upwelling or fluctuating temperatures in shallow reef areas could raise the thermal tolerance of coral, as could the influence of fluctuating environmental factors such as tidal exposure  (Obura, 2005). Although we demonstrated that Symbiodiniaceae mediate P. verweyi acclimatization, we cannot rule out the possibility that mutations to the host itself and natural selection lie behind this species’ ability to adapt to a particular condition. The effect of micro-environment might also explain the difference seen in the Symbiodiniaceae genera association between the native cores and mother colonies. Specificity towards a particular Symbiodiniaceae genus was more apparent in the mother colonies (Fig. 4). Native cores, due to their small size and hence propensity toward stress, showed more flexibility in their associated Symbiodiniaceae genera (Figs. 3A and 3B) (also see the results in Kao et al., 2018).

We hypothesize that combinations of P. verveyi and Symbiodiniaceae genera tend to be specific due to the differences in thermal histories, temperature variations, and hosts favouring one dominant symbiont rather than shuffling. This could have a negative impact on coral exposed to above-threshold thermal anomalies. For example, Kao et al. (2018) observed that, when P. verveyi nubbins were reciprocally transplanted between NPP-OL and WLT, those from WLT that were associated with Cladocopium spp. did not tolerate long-term changes in temperature levels or daily fluctuations. The transplanted nubbins did not survive, even after shuffling. In contrast, nubbins at NPP-OL survived and actually fared well in the more stable and lower-temperature environment of WLT. In fact, they also showed an increase in growth over time, all the while associating with Durusdinium spp., and did not shuffle to Cladocopium spp. even though they could have. Results from this study and symbiont association data and host population genetics at a micro-geographic scale (see Keshavmurthy et al., 2012) hint towards local adaptation in P. verweyi.

Conclusions

Shuffling is not a simple and straightforward way for corals to cope with the effects of climate change, but is in fact a complex process governed by host-symbiont specificity as well as local macro- and micro-environmental conditions. Being flexible (Silverstein, Correa & Baker, 2012) is a good strategy, but specificity is also a norm. In other words, a mere increase in temperature above the threshold limit is not enough to invoke shuffling, even if a coral host has the capacity to associate with multiple symbiont partners (see Kao et al., 2018). While it is popular to be optimistic that shuffling or switching between Symbiodiniaceae genera is a way for corals to survive frequent above-threshold seawater anomalies, we should be cautious, as not all coral species appear to be able to shuffle or switch their associated Symbiodiniaceae genera (e.g., Coffroth et al., 2010), especially corals that have obligate relationships with a particular Symbiodiniaceae genus. We want to reiterate here that, irrespective of corals’ temperature tolerance thresholds in the future, and given the fact that we are facing continuous changes in the global climate through carbon emissions, symbiont shuffling might not be sufficient to withstand frequent and prolonged seawater temperature anomalies as it is not a common trait in all coral species.

Supplemental Information

Supplemental Information 1 QPCR raw data from the native cores collected during 2014 reciprocal transplant experiment

Click here for additional data file.

Supplemental Information 2 QPCR raw data from the native cores collected during 2015 reciprocal transplant experiment

Click here for additional data file.

Supplemental Information 3 QPCR raw data from the tagged mother colonies samples collected during 2014 and 2015 reciprocal transplant experiment

Click here for additional data file.

Supplemental Information 4 Data used to draw figures 4, 5 and 6 of presence/ absence of Durusdinium and Cladocopium

Click here for additional data file.

The authors wish to thank the members of the Coral Lab, Biodiversity Research Center, Academia Sinica (BRCAS; Taipei, Taiwan) for assisting with sampling, field logistics, and molecular analysis. Thanks also to Noah Last of Third Draft Editing for his English language editing.

Additional Information and Declarations

Competing Interests

Author Contributions

Field Study Permissions

Data Availability

The authors declare there are no competing interests.

Shashank Keshavmurthy conceived and designed the experiments, analyzed the data, prepared figures and/or tables, authored or reviewed drafts of the paper, and approved the final draft.

Hwee Sze Tee performed the experiments, analyzed the data, authored or reviewed drafts of the paper, and approved the final draft.

Kuo-Wei Kao conceived and designed the experiments, performed the experiments, analyzed the data, prepared figures and/or tables, and approved the final draft.

Jih-Terng Wang and Chaolun Allen Chen conceived and designed the experiments, authored or reviewed drafts of the paper, and approved the final draft.

The following information was supplied relating to field study approvals (i.e., approving body and any reference numbers):

All the samples used in the experiment were collected through permission granted by the Kenting National Park headquarters (permit numbers 1040008112 and 1040002080) and the Taiwan Power Company for allowing us to work at the Nuclear Power Plant in Kenting.

The following information was supplied regarding data availability:

The raw data are available in the Supplemental Files.

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
