# Peer review of "Specificity trumps flexibility—location-based stable associations between Symbiodiniaceae genera and Platygyra verweyi (Scleractinia; Merulinidae)"

_PeerJ, doi:10.7717/peerj.8791_

## Round 0.1 · original submission · Major Revisions

Two expert reviewers have evaluated your manuscript and their evaluations can be seen below. Both reviewers have made a number of important observations that need to be addressed if you choose to resubmit your manuscript.

Reviewer 1 ·

Basic reporting

-The language is mostly fine, but there are still some typos to check for (e.g. several instances of “form” instead of “from,” and some sentence fragments here and there).

-In most areas the literature is well referenced, but there are a few oversights. For example, the idea that shuffling/switching is taxon-dependent and may not be helpful for the majority of highly specific cnidarians has been around for a while, but there are no references to this early debate (e.g. the Goulet/Baker exchanges in the mid-aughts). There’s also some recent relevant work in anemones showing a lack of shuffling (Gabay et al. 2019 ISMEJ) and trade-offs associated with alternate symbionts (Matthews et al. 2017 PNAS; Matthews et al. 2018 Proc Roy Soc B). There are also opportunities to highlight studies that show a lack of meaningful symbiont shuffling (e.g. Lee et al. 2016 Microbial Ecology) around L374.

-I have a few issues with how Symbiodiniaceae diversity is discussed:

1. The markers used for qPCR are genus level markers, so throughout the manuscript the language should only be about Cladocopium and Durusdinium, rather than Cladocopium C3/C3cc and Durusdinium trenchii/glynni (which implies that those are the only potential species in this association in this location). While other studies may have established that these are the main symbionts in these corals, it’s impossible to rule out that other species may have been present based on the qPCR data, and the language should reflect this.

2. It only seems to be a problem in the introduction, but in L50-55 the authors are presenting an outdated view that different Symbiodiniaceae genera have different, genus-wide physiological characteristics. This is somewhat true for Durusdinium, which generally are found in corals in stressful environments, but it’s wrong to say that “Cladocopium species are the most sensitive to stress.” In fact, the most heat-tolerant symbiont we know of is Cladocopium thermophilum (see studies by Hume et al.). There’s just too much physiological diversity among species within genera to support broad statements about genus-wide characteristics.

3. The writing is unnecessarily vague about what constitutes symbiont switching vs. shuffling (they are distinct phenomena with their own definitions; see e.g. the Baker 2003 review). Relatedly, the writing is also vague about the difference between dominant and background symbionts. In my opinion, just because a coral’s symbiont community changes from 100% Cladocopium to 99% Cladocopium/1% Durusdinium, that doesn’t really constitute shuffling; it’s just an increase in a background symbiont population. Shuffling would require a larger, more ecologically meaningful change. One could argue about the threshold value, but with the exception of the Bay study (which is just one case), I don’t think there’s much compelling evidence to suggest changes < 10% are all that important in most situations. I think the authors are trying to emphasize that certain corals went from entirely C to some mixture of C/D, but that’s not what the language implies. It’s very clear to me from looking at the figures (and the title and conclusions!) that the story should be focused on stability, rather than flexibility, but the wording as is doesn’t really help the case. I think it’s very important to go through the manuscript carefully and make sure this story is clear.

Experimental design

-I feel the manuscript is a bit longer and more complicated than it needs to be. I realize it’s just a portion of a previously published study, but when it comes down to it the reciprocal transplant component is not really necessary to describe in depth (and certainly doesn’t need a figure), as the data are only for parent colonies and smaller fragments of those colonies that were maintained at the same site as the parent. So it’s essentially an observational study of symbiont communities in corals at three locations over several years. That’s not to say it’s not worth publishing, but I think the story could be told far more efficiently.

Validity of the findings

-No comment.

Additional comments

L19: Do you really mean “cores”? Or “corals”? Or “symbiont communities”? This is a confusing opening line and needs a bit more context.

L22: When listing unnamed species, I’d recommend including the entire ITS2 designation (which means adding the former clade letter). So instead of “Cladocopium 3/3cc” I’d write “Cladocopium C3/C3cc.”

L59: Need to define the difference between shuffling and switching.

L80: But what about host mechanisms? Rapid evolution in either partner? I think this line is a bit too narrow.

L108-111: Very confusing sentence. I think you have too many asides. Just split into a few clear sentences.

L171: Which data set? Needs a reference or link.

L299: Here’s the main example of where I think the term “switching” is too strong. I’d rewrite as “In March 2015, cores seemed to have acquired some Cladocopium at various percentages (range: 1-21%; mean: 6%)”

L310: Please provide the mean with the range. Or maybe the median.

L316-318: Long sentence that could be broken into two.

L321: 97% Durusdinium?

L332: Please provide the mean with the range.

L340: typo: Symbiodiniaceae

L341 and 344:

·

Basic reporting

This work looks at the stability of symbionts in one species of coral from Southern Taiwan in colonies located at three different locations. Essentially they observed the presence of two symbiont species. The species from the new genus Cladocopium (formerly Clade C) are found in colonies at locations with environments indicative of the region. All colonies at the third location, the outflow from a nuclear power facility, were dominated by Durusdinium trenchii. This is in keeping with work on other animals from this region of the Pacific. The finding of symbiont stability in these colonies that have not undergone extensive and artificial manipulation is an expected outcome and is likely the rule for most corals everywhere. Still there remain few studies that conduct long-term monitoring and more of these biopsies/long-term monitoring are needed. I think that the authors are entirely correct in that there exists considerable dogma and misinformation about the propensity “switching” and/or “shuffling” and its ecological significance.

I did find the writing of this work a bit convoluted and feel that it could/should be substantially streamlined. They talk about these samples as being part of a larger reciprocal transplant study but then do not show the results of this other work. This is very confusing to the reader. They should simply present it as a yearlong monitoring of colonies in place and leave these other irrelevant and distracting details out (avoid mentioning reciprocal transplants if these were not studied). This also applies to the fitures. Figure 2 should be deleted, Figures 4, 5 and 6 could be easily combined, and Figure 1 modified to remove the reciprocal transplant arrows. With careful editing I think that they could easily reduce the length of this paper by ~50%

I found their taxonomy and taxonomic references of the dinoflagellates symbiont somewhat strange and confusing. For example, the Durusdinium they are tracking in this paper is D. trenchii. This species of symbiont can be found in corals with horizontal symbiont acquisition in Taiwan. Therefore, despite the fact that they are using “genus” level qPCR probes, they should report this Durusdinium as D. trenchii. The only other members of this genus from this region occur in the coral Oulastrea, but these species are specific to that animal.

Moreover, they should remove mention of D. glynnii from the manuscript. This symbiont occurs only in Montiporids and Pocilloporids. The ITS2 sequence variation in both D. trenchii and D. glynnii is very similar and both have one sequence (D1) in common. This I think has created some confusion in their minds
I recommend the authors carefully read the following paper:
Wham, D.C., Ning, G., and LaJeunesse, T.C. (2017). Symbiodinium glynnii sp. nov., a species of stress-tolerant symbiotic dinoflagellates from pocilloporid and montiporid corals in the Pacific Ocean. Phycologia 56, 396-409.

I agree with their general interpretation that the occasional finding of D. trenchii at sites NPP and WLT (and vice versa) is random and that not much emphasis should be placed on these results. Clearly, P. verweyi associates with two species of Symbiodiniaceae and that depending on the location and environment one or the other symbiont can show up (be detected) in a colony.

This brings me to another important issue that needs addressing. It relates to the findings (actually more the conclusions) from the Chen et al. 2005 paper. The perceived seasonality reported by Chen et al. 2005 cannot be substantiated by the present paper nor by a previous follow-up paper conducted by the same group and published in 2012 (Hsu et al. 2012). The findings by Hsu et al. show difference on relative proportions but honestly no seasonality. What appears to be the source of this variability is that many of the colonies in the region have mixtures of Cladocopium and Durusdinium (trenchii) hence sampling artifact at any given time could (and does) create swings in the measured abundances and/or prevalence of a particular species of symbiont. BUT this is not indicative of a predictable ecological response to seasonality. Therefore, I recommend backing off on this in the discussion now and in the future.

It is often mentioned that changing genera occurs withing P. verweyi. This is not accurate. They are working with two specific species from two different genera. While one species is formally described, D. trenchii, unfortunately, the other one is yet to be described. I recommend that they call it Cladocopium sp. C3cc throughout the paper.

The use of the word “shuffling” is also somewhat misleading. Its use implies the invisible hand of the host actively controlling the resident symbiont population, which, to my knowledge, there are no data to support this. I would recommend using more neutral language like “…changes in the relative proportions of symbiont occurred…”

With regard to their discussion on stability, the authors also should read (and cite) a paper on a similar story showing prevalent stability in a high latitude system/coral species:
Lee, M.J., Jeong, H.J., Jang, S.H., Lee, S.Y., Kang, N.S., Lee, K.H., Kim, H.S., Wham, D.C., and LaJeunesse, T.C. (2016). Most Low-Abundance "Background" Symbiodinium spp. Are Transitory and Have Minimal Functional Significance for Symbiotic Corals. Microbial Ecology.

There is mention of NGS but this approach is plagued with its own problems, so if they’ve been criticized previously for not using this, I think that this was unfair.


Be careful when reporting the interpreted the findings of the Boulotte paper. This paper is plagued with inconsistencies and some of the data are rather peculiar. To my knowledge and I have worked on these animals across the Indo-Pacific, P. damicornis and Stylophora do not associate with the same symbiont species, yet this work shows the same symbiont in both species of host. See Sampayo et al. 2007 for example of the marked differences in the symbionts harbored by these two Pocilloporidae.

I disagree with the assertion, “Studies have shown that corals that are able to shuffle have a better chance of overcoming stress,” there is very little evidence to support this speculation, further changes in symbiont are more of a sign of severe stress on the animal than an acclamatory mechanism.

Experimental design

see above

Validity of the findings

see above

Additional comments

see above

---

## Round 0.2 · Minor Revisions

I am satisfied with the scientific changes made to the manuscript.

However, there are some issues with the use of English, which should be corrected using an English Language Service or by a native English speaker.

---

## Round 0.3 · accepted · Accept

I am satisfied with the editorial changes made to the manuscript.